# Regulation of the SUV39H Family Methyltransferases: Insights from Fission Yeast

**DOI:** 10.3390/biom13040593

**Published:** 2023-03-25

**Authors:** Rinko Nakamura, Jun-ichi Nakayama

**Affiliations:** 1Division of Chromatin Regulation, National Institute for Basic Biology, Okazaki 444-8585, Japan; 2Department of Basic Biology, School of Life Science, The Graduate University for Advanced Studies, SOKENDAI, Okazaki 444-8585, Japan

**Keywords:** histone methylation, histone methyltransferases, SUV39H, Clr4, fission yeast, chromodomain, SET domain, autoregulation, ubiquitylation

## Abstract

Histones, which make up nucleosomes, undergo various post-translational modifications, such as acetylation, methylation, phosphorylation, and ubiquitylation. In particular, histone methylation serves different cellular functions depending on the location of the amino acid residue undergoing modification, and is tightly regulated by the antagonistic action of histone methyltransferases and demethylases. The SUV39H family of histone methyltransferases (HMTases) are evolutionarily conserved from fission yeast to humans and play an important role in the formation of higher-order chromatin structures called heterochromatin. The SUV39H family HMTases catalyzes the methylation of histone H3 lysine 9 (H3K9), and this modification serves as a binding site for heterochromatin protein 1 (HP1) to form a higher-order chromatin structure. While the regulatory mechanism of this family of enzymes has been extensively studied in various model organisms, Clr4, a fission yeast homologue, has made an important contribution. In this review, we focus on the regulatory mechanisms of the SUV39H family of proteins, in particular, the molecular mechanisms revealed by the studies of the fission yeast Clr4, and discuss their generality in comparison to other HMTases.

## 1. Introduction

In eukaryotic cells, DNA is tightly compacted as chromatin, and dynamic changes in chromatin structure have been linked to transcriptional regulation. The basic unit of chromatin is the nucleosome; the histone octamer containing two copies each of histones H2A, H2B, H3, and H4, forms the core, around which ∼147 bp of DNA wraps. The N-terminal tails of histones are subject to post-translational modifications (PTMs), such as acetylation, methylation, phosphorylation, and ubiquitylation [1], and these histone tail PTMs play key roles in the transcriptional regulation and dynamic changes in chromatin structure. Among these, histone methylation is one of the most important PTMs on histones, as it serves different cellular functions depending on the location of the amino acid residue undergoing modification.

The existence of histone methyl modifications has been known since the 1960s [2], but the enzymes responsible for methylation had been elusive for a long time. In 2000, members of the SUV39H family of proteins were identified as the first histone methyltransferases (HMTases), followed by the identification of a large number of HMTases in different species based on information from the evolutionarily conserved catalytic domain. The subsequent identification of a number of histone demethylases that enzymatically remove this methyl modification supported the idea that histone methylation is a modification that, like acetylation and phosphorylation, is regulated by antagonistic enzymatic reactions. Further studies of HMTases in various model organisms have shown that the activity of HMTases, including those of the SUV39H family, is regulated at different levels.

In this review, we will focus on the SUV39H family of HMTases, in particular, the fission yeast Clr4; briefly introduce the background of their discovery; and discuss how the activity of this family of enzymes is regulated. We will also discuss whether the regulatory mechanisms found in this family are common to other HMTases.

## 2. Identification of the SUV39H Family of HMTases

Heterochromatin is a highly condensed chromatin structure observed in the nucleus. Heterochromatin can be divided into two major types: one is called constitutive heterochromatin, which is always condensed; rich in non-coding, highly repetitive sequences; and primarily associated with centromeres and telomeres. The other is called facultative heterochromatin, in which genomic regions, including genes, are inactivated during development. A well-known example of the facultative heterochromatin is the inactivated X chromosome.

In *Drosophila*, chromosomal rearrangements can be induced, for example, by ionizing radiation, and when genes originally located in euchromatin are placed in the vicinity of pericentromeric heterochromatin due to chromosomal rearrangements, the expression of the euchromatic genes can become variegated. This phenomenon, called position effect variegation (PEV), is thought to be caused by heterochromatin spreading into the euchromatic gene region and has become a very useful experimental system for genetic and molecular studies of heterochromatin. For example, if a mutation has a suppressor effect on PEV, the gene could encode a critical component of heterochromatin, and if a mutation has an enhancer effect on PEV, the gene could encode a product associated with chromatin decondensation. Using this criterion, genetic analysis of suppressors and enhancers of PEV revealed >100 genes closely associated with the regulation of PEV. These genes were collectively named *Su(var)* and *E (var)* genes. One of the suppressor genes, designated *Su(var)3-9*, was cloned and found to encode a protein with two characteristic domains, the chromo and SET domains [3]. Based on the sequence information of the SET domain, human (*SUV39H1*) and mouse (*Suv39h1*) homologues were isolated [4]. It should also be noted that another suppressor gene, called *Su(var)2-5*, has been shown to encode heterochromatin protein 1 (HP1) [5].

In fission yeast, the mating-type loci contain three cassettes called *mat1*, *mat2-P*, and *mat3-M*. The mating-type of each cell, P or M, is determined by the gene placed at the *mat1* locus, and both the *mat2-P* and *mat3-M* loci are transcriptionally silent and serve only to switch the mating-type cassette at the *mat1* locus. A genetic screen to isolate mutants that derepress the silent mating-type loci identified several genes, including the *cryptic loci regulator 4* (*clr4*), which could encode transacting factors involved in the silencing of the mating-type loci [6]. The *clr4* gene was also shown to be essential for centromere silencing [7]. The *clr4* gene was cloned by functional complementation and found to encode a protein with high homology to the product of *Su(var)3-9* in *Drosophila* [8].

The chromodomain (CD) was originally identified as a protein sequence motif common to the *Drosophila* Polycomb (Pc) and HP1 and shown to be required for the chromatin targeting of these proteins [9,10]. The evolutionarily conserved SET domain was initially characterized as a common motif in the PEV modifier Su(var)3-9, the Polycomb-group protein E(z) (encoded by the *enhancer of zest* [*E(z)*]), and the Trithorax-group protein TRX [3]. While the detailed function of these domains was not readily apparent, the SET domain of Su(var)3-9 was found to have significant sequence similarity to plant methyltransferases, and this knowledge was used to identify the SUV39H family of proteins as histone H3 lysine 9-specific methyltransferases [11]. This discovery led to the identification of many other SET domain proteins as histone methyltransferases with different substrate specificities. In addition, the chromodomain found in HP1 and PC was shown to be a domain that specifically recognizes methylated histones [12,13,14].

Although the SET domain is now widely recognized as the catalytic domain of HMTases, SET domain proteins with non-histone protein substrates appear to be more common, as their initial discovery was triggered by the plant RUBISCO methyltransferase. For example, the fission yeast genome encodes 13 SET domain proteins, at least 3 of which have been shown to specifically methylate ribosomal proteins [15,16]. Given that the methylation of some ribosomal proteins is also conserved in prokaryotes, such as *Escherichia coli*, the SET domain-containing methyltransferases may have evolved to modulate target proteins rich in basic residues that bind to DNA or RNA.

## 3. Mechanisms Regulating the Activity of SUV39H Family of HMTases

### 3.1. Structure of the SUV39H Family of HMTases

The SUV39H family of HMTases possess two evolutionarily conserved functional domains: the N-terminal chromodomain (CD) and the C-terminal SET domain. The SET domain is flanked by N-SET, pre-SET, and post-SET domains and contains a SET insertion (SET-I) domain, which has been proposed to play a role in substrate recognition and cofactor binding. The SET domain has catalytic activity to methylate lysine 9 of histone H3 (H3K9), and the CD specifically binds the K9 methylated histone H3 tail. As described in the later section (Section 3.3), this read/write mechanism is coupled to the regulation of the enzymatic activity and is required for the maintenance and spreading of heterochromatin. The CD and the SET domain are connected by a disordered region. In addition, some members of the SUV39H family of proteins possess a disordered region of variable length at the N-terminus (in this manuscript, we will refer to the disordered region connecting the CD and the SET domain as the “hinge region” to distinguish it from the unstructured region at the N-terminus) (Figure 1). In contrast to the CD and the SET domain, the hinge and N-terminal disordered regions are less evolutionarily conserved, vary in length, and their functional roles are not well understood. However, as discussed below, they may be involved in their specific functions. For example, mammalian cells express two SUV39H family proteins, Suv39h1 and Suv39h2, and Suv39h2 has a basic domain in its N-terminal disordered region that is not found in Suv39h1. As described below, this basic domain is involved in binding to RNA and in the targeting of Suv39h2 to heterochromatin [17]. In *Neurospora crassa*, a SET domain protein, DIM-5, catalyzes H3K9 methylation and is thought to be a functional homologue of the SUV39H family of HMTases [18]. Interestingly, no obvious CD-like structure is found at the N-terminal side of the SET domain in DIM-5. The role of CD in other SUV39H family HMTases may be carried by other interacting factors. In the basidiomycete yeast *Cryptococcus neoformans*, the Clr4 homolog does not contain any characteristic structure other than the SET domain, and it has a long-disordered region consisting of about 1,400 amino acid residues on the N-terminal side of the SET domain [19]. It will be interesting to investigate how these disordered regions are involved in the function of each protein.

### 3.2. Autoinhibition

Like many other enzymes, the activity of HMTases is thought to be tightly regulated. In particular, if the SUV39H family of HMTases is dysregulated and repressive H3K9me marks are placed throughout the genome, this would result in the repression of genes essential for cell growth or other cellular functions. An elegant study in fission yeast revealed a mechanism that regulates Clr4 HMTase activity.

In vitro experiments with recombinant Clr4 showed that Clr4 has the activity to methylate itself, and mutation experiments revealed that lysine residues K455 and K472 are the primary targets of this automethylation. Interestingly, X-ray crystallographic analysis revealed that these lysine residues are located in a loop (amino acids 453–472, termed the “autoregulatory loop”) connecting the SET and post-SET domains (Figure 1). Further biochemical and structural experiments revealed that the loop inhibits the Clr4′s catalytic activity by blocking the histone H3K9 substrate-binding pocket, and that automethylation of specific lysine residues in this loop promotes a conformational switch. The importance of this automethylation switch was further confirmed by in vitro and in vivo experiments: Clr4 with lysine-to-arginine substitutions, which mimic lysine residues that cannot be methylated by Clr4, showed hypoactivity in vitro and attenuated heterochromatin silencing in vivo, whereas Clr4 with lysine-to-alanine (K-to-A) substitutions, which contain a small non-polar side chain and block the interaction between the autoregulatory loop and the active site, showed hyperactivity in vitro and enhanced heterochromatin silencing. These results suggest that the automethylation-induced conformational switch in Clr4 plays a critical role in regulating H3K9me to maintain the heterochromatin structure (Figure 2A) [20].

One of the most important questions is whether this automethylation-induced conformational switch is an evolutionarily conserved mechanism. In the studies described above, two lysine residues, K455 and K472, were identified and analyzed as major self-methylation sites, but Clr4 K455 is not conserved in three related *Schizosaccharomyces* species (*S. cryophilus*, *S. octosporus*, and *S. japonicus*); the corresponding amino acid residue is arginine instead of lysine. Since Clr4 K472 is conserved in these related species and is located at the end of the autoregulatory loop, it is possible that automethylation of K472 is primarily responsible for the conformational switch.

In the crystal structure of human SUV39H2, a similar loop structure and lysine residues, K375 and K392, corresponding to Clr4 K455 and K472 are found between the SET and post-SET domains [21]. A recent study reported that human SUV39H2 is subject to automethylation at K392 [22]. These observations suggest that the mechanism is evolutionarily conserved. However, it should be noted that automethylation of human SUV39H2 blocks the substrate interaction and negatively regulates the methyltransferase activity [22]. Further analysis is required to determine whether the automethylation of the SUV39H family of HMTases and their regulatory mechanisms are evolutionarily conserved. Related to the automethylation-induced conformational switch, a recent report showed that Clr4 S458, located in the autoregulatory loop, is differentially phosphorylated by the cyclin-dependent kinase Cdk1 during meiosis and that the phosphorylation states are accompanied by the changes from H3K9me2 to H3K9me3 [23]. It would be interesting to determine how S458 phosphorylation modulates Clr4′s activity and the automethylation-induced conformational switch.

Polycomb-repressive complex 2 (PRC2) plays a critical role in regulating transcriptional repression in mammals. EZH2, a catalytic subunit of the PRC2 complex, is responsible for the trimethylation of H3K27 (H3K27me3) and also undergoes automethylation. A recent study has shown that three lysine residues (K510, K514, and K515) located in a disordered but evolutionarily conserved loop in the SET domain are methylated. Methylation of these lysine residues increases PRC2 methyltransferase activity, whereas their mutation (K-to-A) decreases activity in vitro. Experiments with cell lines expressing mutant EZH2 have shown that EZH2 automethylation is critical for H3K27me3 catalysis [24,25]. Although the effect of the K-to-A mutation on in vitro HMTase activity was opposite to that of Clr4, it is noteworthy that, in both cases, automethylation occurs on an internal disordered loop, and methylation increases HMTase activity. It is likely that the flexible loop containing critical lysine residues acts as a pseudosubstrate, blocking its enzymatic activity, and that when the H3 tail or cofactor (SAM) become available, it is released from the substrate-binding domain to methylate the substrates. Since other SET domain-containing HMTases have also been reported to have automethylation activity [26,27], it will be interesting to see if the automethylation-induced activation is a common mechanism in other HMTases.

### 3.3. Substrate Recognition

When HMTases, including SUV39H1 family proteins, methylate H3K9 on nucleosomes, there can be several patterns. One is when H3K9 methylation is catalyzed in chromatin regions where there is no pre-existing H3K9me, such as when a transposon or other foreign genetic element is newly inserted into a chromosomal region. In such cases, H3K9 methylation may be introduced de novo through the regulation of methyltransferase activity in concert with pathways, such as RNA silencing, as has been extensively studied in fission yeast. The second case is when nucleosomes containing H3K9 methylation and nucleosomes with unmodified H3K9 are intermingled or adjacent, as may occur when chromosomal regions containing H3K9 methylation are replicated in the S phase of the cell cycle or when heterochromatin regions are spread to neighboring chromatin regions. As mentioned above, the SUV39H family methyltransferases have CD at the N-terminus, and in the latter case, the recognition of H3K9me by CD is expected to be coupled with methylation activity. This mechanism has been proposed as the read/write model. Extensive biochemical experiments have shown that the activity of SUV39H family HMTases is indeed regulated by such a read/write mechanism.

When dinucleosomes, synthesized by ligating a nucleosome containing unmodified H3K9 and a nucleosome containing H3K9me3, were used as substrates, it was found that Clr4 exhibited several-fold higher activity compared to control dinucleosomes and that the effect of H3Kme3 is dependent on a functional Clr4 CD. The addition of mononucleosomes containing H3K9me3 to the same reaction did not show a positive effect on the Clr4 HMTase activity, suggesting that this is not a simple allosteric effect of Clr4, but a cis effect that occurs in the case of adjacent nucleosomes (Figure 2B) [28]. This study also suggests that the CD of Clr4 is more specific for H3K9me3 than that of other HP1 proteins (Swi6 and Chp2), contributing to the division labor among CD proteins.

Similar experiments were performed on human SUV39H1, in which the activity of SUV39H1 was examined using trinucleosome substrates synthesized by ligating H3K9me3-containing nucleosomes and H3K9-unmodified nucleosomes in various combinations. As shown for Clr4, SUV39H1 shows a stronger HMTase activity in the presence of the adjacent H3K9me3 nucleosome. Interestingly, SUV39H1 HMTase activity was also enhanced by the addition of H3K9me3 peptide, suggesting that SUV39H1 HMTase activity is allosterically controlled [29]. The reason why the allosteric effect differed between Clr4 and SUV39H1 is unclear, but the H3K9me3 peptide may have been more effective than mononucleosomes containing H3K9me3. In any case, these studies have confirmed that the binding of the N-terminal CD to H3K9me3 promotes the HMTase activity of both Clr4 and SUV39H, but how do the CDs not bound to H3K9me3 inhibit the methyltransferase activity? The simple mechanism would be that the N-terminal region containing the CD interacts with the C-terminal catalytic domain or its substrate binding site and inhibits activity. This idea would be tested by further detailed biochemical analysis. Since a similar read/write mechanism must exist for other HMTases that do not have the CD recognizing methylated histones in the same molecule, it would also be very interesting to see how this works in cooperation with other interacting factors.

### 3.4. RNA Biniding

As mentioned above, CD is the domain that recognizes H3K9me3, suggesting that it contributes to the read/write mechanism of SUV39H family histone methyltransferases, but the flanking regions of CD have been shown to have roles other than H3K9me3 recognition. In fission yeast, repetitive sequences in the pericentromeric region are transcribed by RNA polymerase II, and the transcribed RNA is converted to double-stranded RNA by RNA-dependent RNA polymerase (Rdp1), which is cleaved by Dicer (Dcr1) to form small RNA molecules (siRNAs). The siRNA is then incorporated into an effector complex, RNA-induced transcriptional silencing (RITS), which targets heterochromatin to promote Clr4 to deposit H3K9me3. This transcription by RNA polymerase II and the introduction of H3K9me3 by Clr4 are coupled, and the whole process is regulated as a feedback loop. Recognition of nascent RNA by the RITS complex is primarily accomplished by base pairing between siRNAs in RITS and transcribed nascent RNA, but it has been suggested that sequence-independent RNA binding by other factors also contributes to the targeting of RITS to the heterochromatin region. For example, Chp1, a component of the RITS complex, has a CD, but there is a cluster of basic amino acid residues in the C-terminal α-helix region, and it has been shown that Chp1 binds to RNA through this CD-flanking region and that the RNA-binding ability of Chp1 is necessary for Chp1 function [30]. Interestingly, there is a similar cluster of basic amino acid residues in the Clr4 CD, and this region has RNA-binding activity, and mutations in this region result in abnormal Clr4 function (Figure 2C) [30]. It is not well understood how the RNA-binding ability of the Clr4 CD is involved in the regulation of Clr4 HMTase activity, but it may be that the recognition of H3K9me3 by the CD and the RNA binding by the flanking region of CD work in concert to promote Clr4 HMTase activity.

Do other members of the SUV39H family of HMTases have similar RNA-binding activity, and is such activity involved in their function? While no obvious RNA silencing mechanism is known to be involved in the formation of pericentromeric heterochromatin in human and mouse cells, as has been demonstrated in fission yeast, it has been reported that satellite repeats are transcribed at the pericentromeric regions [31]. The association between human and mouse SUV39H proteins and RNA-binding ability was examined. Both human and mouse SUV39H1 bind to RNA via basic amino acid residues in the CD, and both H3K9me3 recognition and RNA-binding ability, both associated with the CD, are important for SUV39H1 function at pericentromeric heterochromatin [32,33]. Interestingly, in a functional rescue experiment using Suv39h1 and Suv39h2 double-knockout ES cells, the RNA-binding ability, rather than the H3K9me3 recognition by CD, was critical for H3K9me3 recovery [32]. The RNA-binding activity of SUV39H1 may play an important role in introducing H3K9me3 de novo to chromosomal regions where H3K9me3 is absent.

Human and mouse cells express SUV39H2, in addition to SUV39H1, and SUV39H2 has a basic domain enriched in basic amino acid residues at its N-terminal region. This basic domain plays an important role in Suv39h2 heterochromatin binding, and detailed biochemical analyses showed that, like Suv39h1, Suv39h2 binds to single-stranded RNAs derived from major satellite repeats via this basic domain and that Suv39h2 chromatin-binding is RNA-dependent [17]. Interestingly, major satellite repeat RNAs have been shown to form DNA/RNA hybrids in cells, suggesting that Suv39h1 and Suv39h2 may target higher-order nucleosome structures containing RNA. Thus, using fission yeast and mammalian cells, it has been shown that RNA-binding capacity is involved in the function of SUV39H family HMTases, but whether this is a mechanism common to the SUV39H family remains to be determined. Similar to the mammalian Suv39h2, *Drosophila* Su(var)3-9 contains the N-terminal disordered region, which is rich in basic amino acid residues, suggesting that the RNA-binding ability associated with this region may contribute to the targeting of Su(var)3-9 to heterochromatin. Notably, the PRC2 complex containing EZH2 also binds to RNA, but the presence of RNA inhibits H3K27 methylation activity by preventing the binding of the substrate nucleosome [34,35,36]. Therefore, it will be a future challenge to elucidate how the RNA-binding activity found in the SUV39H family proteins and their recognition of substrate nucleosomes, including DNA, are coordinated in the chromatin context of the cells.

### 3.5. Crosstalk of Histone Modifications

In fission yeast, Clr4 forms a protein complex called the Clr4 methyltransferase complex (CLRC). The CLRC consists of the cullin scaffold protein Cul4, the β-propeller protein Rik1, the WD-40 protein Raf1 (Dos1/Cmc1/Clr8), the replication foci targeting sequence (RFTS)-like domain-containing protein Raf2 (Dos2/Cmc2/Clr7), and the RING-box protein Rbx1 (Pip1), and these CLRC components are all required for heterochromatin silencing [37,38,39,40,41]. Cul4, Rik1, and Raf1 show a structural similarity to the conserved CUL4-DDB1-DDB2 E3 ubiquitin ligase (CRL4^DDB1^) [42,43]. Although no Raf2 equivalent component is found in the CRL4^DDB1^ complex, Raf2 interacts with Cul4, Rik1, and Raf1, suggesting that Raf2 acts as a hub in the CLRC complex [43,44]. As expected from its structural similarity to CRL4^DDB1^, CLRC exhibits ubiquitylating activity in vitro [37]. However, whether the CLRC acts as an E3 ubiquitin ligase in vivo and how ubiquitylation modulates the HMTase activity of Clr4 remained unclear.

Previously, we performed an in vitro ubiquitylation assay using affinity-purified CLRC and demonstrated that CLRC preferentially ubiquitylates lysine 14 on histone H3 (H3K14). Mass spectrometry analysis of chromatin immunoprecipitated fractions revealed that the H3K14 ubiquitylation (H3K14ub) is tightly associated with H3K9me-enriched heterochromatin in vivo. Importantly, by using K14-ubiquitylated histone H3 as a substrate in the in vitro methyltransferase assay, we further demonstrated that the K14-ubiquitylated H3 promotes the methyltransferase activity of Clr4 [45]. The promotion of Clr4 activity by H3K14ub was confirmed by subsequent sophisticated biochemical experiments, and further structural analysis identified a region within the catalytic domain of Clr4 that directly interacts with ubiquitin [46]. Taken together, these studies demonstrated a crosstalk mechanism between histone methylation and ubiquitylation for heterochromatin assembly (Figure 3).

Previous studies have shown that the H3K14-specific histone deacetylase Clr3 is important for heterochromatin formation in fission yeast [14,47,48], but it was unclear how H3K14 acetylation (H3K14ac) regulates Clr4 activity, because H3K14ac does not directly block Clr4 activity [14]. The finding that Clr4 activity is promoted by H3K14ub suggests that Clr4 activity is regulated by a stepwise mechanism, in which deacetylation by Clr3 occurs first, CLRC ubiquitylates H3K14 by targeting unmodified H3K14, and H3K14ub promotes Clr4 activity to methylate H3K9 (Figure 3). Judging from the mass spectrometry data, the abundance of H3K14ub is low, suggesting that H3K14ub is a transient modification that may be dynamically regulated by the involvement of deubiquitylating enzymes, possibly in conjunction with the cell cycle.

In *Neurospora crassa*, the H3K9 methyltransferase Dim-5 forms a complex with Cul4 and DDB1 called the DCDC, which is required for Dim-5-mediated H3K9me [49]. In human cells, CUL4 is associated with SUV39H1 [50], and depletion of CUL4 or DDB1 impairs H3K9me3 [51]. In addition, CUL4-DDB1 has been shown to have ubiquitin ligase activity for H3 [52], and proteomic studies using human cells revealed that H3K14 is ubiquitylated in vivo [53]. It is, therefore, likely that CUL4-DDB1-mediated H3 ubiquitylation plays an important role in the methylation of H3K9 in other systems. In support of this idea, H3K14ub promotes the activity of human SUV39H1 and SUV39H2 in an in vitro methylation assay, as demonstrated for Clr4 [46] (our unpublished observations). Whether H3K14ub is involved in heterochromatin formation in human cells and whether CRL4^DDB1^ or an unidentified E3 ligase is involved in the ubiquitylation of H3 in heterochromatin will be the subject of future studies.

Crosstalk between histone methylation and ubiquitylation has been demonstrated for other HMTases. For example, the activity of budding yeast Set1, an H3K4 HMTase, requires H2B monoubiquitylation [54,55,56,57], and histone H2A monoubiquitylation promotes H3K27 methylation by PRC2 [58]. Beyond the HMTases, histone H3 ubiquitylation mediated by UHRF1 promotes the activity of DNMT1 [59,60], one of the mammalian DNA methyltransferases. Analysis of the dynamics of intracellular ubiquitin modifications will likely lead to further understanding of the crosstalk between ubiquitylation and methylation.

## 4. Summary and Perspective

In this review, we focus on the regulatory mechanisms of the activity of the SUV39H family enzymes, in particular the results obtained in the study of Clr4 in fission yeast. As mentioned above, the activity of the SUV39H family is regulated in several steps: inhibition of enzymatic activity by automethylation, promotion of enzymatic activity by CD’s H3K9me3 recognition on the neighboring nucleosomes, targeting to heterochromatin by intrinsic RNA-binding activity, and crosstalk between histone ubiquitylation and methylation. We hope you will understand that some of these regulatory mechanisms have been revealed, mainly through the analysis of Clr4 in fission yeast. While the details of the individual regulatory mechanisms have been elucidated, and we have discussed their evolutionary conservation, but the key question that remains unanswered is how these regulatory mechanisms are integrated within the cell to control the activity of the enzyme. How is the automethylation of Clr4 and other SUV39H family HMTases regulated in cells? How can Clr4 and SUV39h family HMTases act on target chromatin in the absence of H3K9me3, and how is CD-mediated repression released? How is the RNA-binding activity of Clr4 and SUV39H family HMTases regulated in a way that does not compete with binding to nucleosomal DNA? How is histone H3 ubiquitylation dynamically regulated in the cell? These important questions should be addressed in the future by integrating our knowledge of SUV39H family enzymes from different species.

## Figures and Tables

**Figure 1 biomolecules-13-00593-f001:**
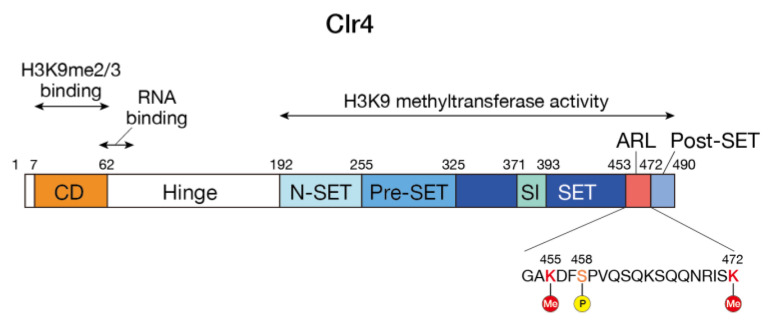
Domain organization of *S. pombe* Clr4 and the positions of automethylated lysine (K455 and K472) and serine (S458) phosphorylated by Cdk1. ARL, autoregulatory loop; CD, chromodomain; SI, SET insertion domain.

**Figure 2 biomolecules-13-00593-f002:**
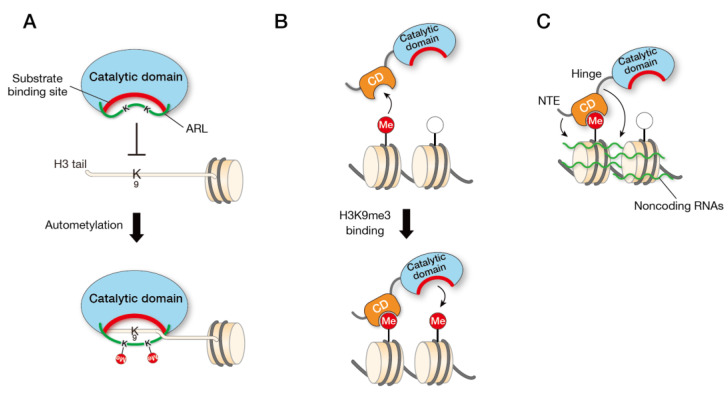
Regulatory mechanisms of Clr4/SUV39H HMTases. (**A**) Model for Clr4 autoregulation by automethylation of two lysines in Clr4. In the unmethylated state of lysines in the Clr4 autoregulatory loop (ARL), the ARL blocks the substrate binding and the methyltransferase activity of Clr4 is inhibited. Methylation of lysine residues within the ARL changes the conformation of the ARL, allowing the Clr4 catalytic domain to bind the H3 tail. (**B**) Model for a two-step activation of Clr4/SUV39H methyltransferase activity. In the absence of Clr4/SUV39H-CD binding to H3K9me, the methyltransferase activity is suppressed. When the CD of Clr4/SUV39H binds to the H3K9me of a nucleosome, the binding of Clr4/SUV39H to chromatin is stabilized, allowing methylation of H3K9 of adjacent nucleosomes. (**C**) Model for chromatin tethering of Clr4/SUV39H1 through RNA-binding activity. H3K9me3 modifications of histones and non-coding RNAs transcribed from pericentromeric repeats work together to promote the association of Clr4/SUV39H with heterochromatin. NTE, N-terminal extension.

**Figure 3 biomolecules-13-00593-f003:**
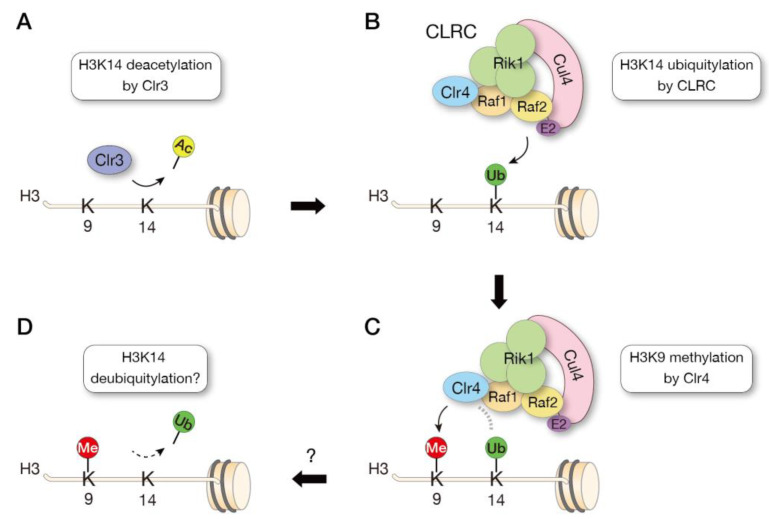
Model of crosstalk between histone methylation and ubiquitylation. The fission yeast histone methyltransferase Clr4 forms a CLRC complex with factors involved in ubiquitylation. Heterochromatin formation is thought to occur in the following steps: deacetylation of H3K14 by Clr3 (**A**), ubiquitylation of H3K14 by the CLRC complex (H3K14ub) (**B**), and methylation of H3K9 by Clr4 (**C**). Clr4 activity is promoted by the presence of H3K14ub. H3K14ub is thought to be subsequently removed by deubiquitylating enzymes (**D**), but the catalytic enzymes and their regulatory mechanisms remain to be elucidated.

## Data Availability

Not applicable.

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
