# Peer review of "Regulation of the SUV39H Family Methyltransferases: Insights from Fission Yeast"

_biomolecules, 2023, doi:10.3390/biom13040593_

Round 1

Reviewer 1 Report

This review offers a comprehensive understanding of the current knowledge on SUV39H family methyltransferases (MTases), highlighting their role in histone H3K9 methylation, chromatin condensation, and transcription repression. The authors delve into the identification of SUV39H MTases and explore structural and biochemical studies of Clr4 and SUV39H, examining their interactions with RNA and other histone modifications. This review provides valuable insights into how SUV39H MTases self-regulate and interact with chromatin, RNA, and other histone modifications, collectively contributing to their function in chromatin condensation.

However, it is important to note that there are certain statements that need revising before publishing in Biomolecules.

Major points

1.     Lane 158-161: the notion that lysine-to-arginine substitution mimics an unmethylated state and lysine-to-alanine substitution mimics a methylated state is incorrect, as both substitutions disrupt lysine methylation, and it should be noted that arginine can also undergo methylation.

2.     Lane 227: the use of the term "de novo" when referring to trimethylation of H3K27 may not be appropriate, as it is unclear whether this modification is always established through a de novo pathway.

Author Response

Referee #1

This review offers a comprehensive understanding of the current knowledge on SUV39H family methyltransferases (MTases), highlighting their role in histone H3K9 methylation, chromatin condensation, and transcription repression. The authors delve into the identification of SUV39H MTases and explore structural and biochemical studies of Clr4 and SUV39H, examining their interactions with RNA and other histone modifications. This review provides valuable insights into how SUV39H MTases self-regulate and interact with chromatin, RNA, and other histone modifications, collectively contributing to their function in chromatin condensation.

Response: We thank the reviewer for his/her positive comments on our manuscript.

However, it is important to note that there are certain statements that need revising before publishing in Biomolecules.

Major points

  1. Lane 158-161: the notion that lysine-to-arginine substitution mimics an unmethylated state and lysine-to-alanine substitution mimics a methylated state is incorrect, as both substitutions disrupt lysine methylation, and it should be noted that arginine can also undergo methylation.

Response: We thank the reviewer for raising the important point and apologize for our inappropriate description. As suggested by the reviewer, we have revised our manuscript to explain the exact purpose of amino acid substitutions as follows:

>3.2 Automethylation (line 181-185): Clr4 with lysine-to-arginine substitutions, which mimic lysine residues that cannot be methylated by Clr4, showed hypoactivity in vitro and attenuated heterochromatin silencing in vivo, whereas Clr4 with lysine-to-alanine (K-to-A) substitutions, which contain a small non-polar side chain and block the interaction between the autoregulatory loop and the active site, showed hyperactivity in vitro and enhanced heterochromatin silencing.

Response: Although the reviewer also suggested to note that arginine can also be methylated, it seems unlikely that the substituted arginine in the autoregulatory loop is targeted by other cellular arginine methyltransferases. Moreover, since this review article mainly focuses on the SUV39H-family lysine methyltransferases, mentioning arginine methylation may confuse the readers. Therefore, we decided that it would be better not to describe arginine methylation to avoid confusion.

  1. Lane 227: the use of the term "de novo" when referring to trimethylation of H3K27 may not be appropriate, as it is unclear whether this modification is always established through a de novo pathway.

Response: The reason that we described "de novo" in the relevant section is that in the paper by Wang et al. (Genes & Dev, 33, 1416-1427, 2019, new reference #24), the authors used EZH2-depleted HEK293T cells and showed that WT but not methyl-defective mutant EZH2 could rescue the deposition of H3K27me3. Based on this result, we describe the EZH2 activity as de novo, as described in the paper. However, in the EZH2-depleted HEK293T cells, H3K27me3 is not completely gone, but there is residual H3K27me3. Therefore, the possibility that EZH2 activity is affected by the existing H3K27me3 cannot be ruled out, so we refrain from using “de novo” and changed the text as follows.

3.2 Autoinhibition (line 256-257) > Experiments with cell lines expressing mutant EZH2 have shown that EZH2 automethylation is critical for H3K27me3 catalysis.

Reviewer 2 Report

This current manuscript is a timely and comprehensive review of the regulatory mechanisms of the SUV39H family of histone methyltransferases (HMTases), particularly those identified in the course of the study of Clr4 protein, a yeast homolog of the mammalian SUV39H1 and SUV39H2 enzymes.

The authors first described a brief history of the identification of SUV39H family, clarifying the current nomenclature of these enzymes. This is a good introduction to the main part of the manuscript devoted to the mechanisms regulating the activity of the enzymes in various evolutionary different species. The authors discussed there in more detail the structure of the enzyme, the mechanisms behind substrate recognition and targeting to heterochromatin, and the importance of the enzyme automethylation for the regulation of its activity. Finally, the authors briefly discussed the role of Clr4 in the crosstalk between histone methylation and ubiquitination.

Comments: In general, I found this review to be well structured and documented and it is also easy to understand its meaning. The figures are simple and clear and do not require too much time for interpretation. However, I have a problem with the sentence ‘The existence of histone methyl modifications has been known as the 1980s’ (line 37), since this information is misleading. It was established as early as the late 1960s that a lysine NH2 group in histone was monomethyl, dimethyl or trimethyl (for a review, see PMID: 17291768). The authors should correct the sentence and provide an appropriate citation.

In summary, this is a timely and interesting review that is worth publishing in Biomolecules after minor revision.

Author Response

Referee #2

Comments: In general, I found this review to be well structured and documented and it is also easy to understand its meaning. The figures are simple and clear and do not require too much time for interpretation.

Response: We thank the reviewer for the positive comments on our manuscript.

However, I have a problem with the sentence ‘The existence of histone methyl modifications has been known as the 1980s’ (line 37), since this information is misleading. It was established as early as the late 1960s that a lysine NH2 group in histone was monomethyl, dimethyl or trimethyl (for a review, see PMID: 17291768). The authors should correct the sentence and provide an appropriate citation.

Response: We thank you for your suggestions and apologize for our oversight. We have modified the text as follows and added an appropriate reference.

1. Introduction (line 37-38) >The existence of histone methyl modifications has been known since the 1960s [2], but the enzymes responsible for methylation had been elusive for long time.

New reference:

2. Murray, K. The Occurrence of Epsilon-N-Methyl Lysine in Histones. Biochemistry 1964, 3, 10-15, doi:10.1021/bi00889a003.